# Transient Receptor Potential Vanilloid 4-Dependent Microglial Function in Myelin Injury and Repair

**DOI:** 10.3390/ijms242317097

**Published:** 2023-12-04

**Authors:** Jameson P. Holloman, Sophia H. Dimas, Angela S. Archambault, Fabia Filipello, Lixia Du, Jing Feng, Yonghui Zhao, Bryan Bollman, Laura Piccio, Andrew J. Steelman, Hongzhen Hu, Gregory F. Wu

**Affiliations:** 1Department of Neurology, Washington University School of Medicine, Saint Louis, MO 63110, USAfilipellof@wustl.edu (F.F.);; 2Department of Animal Sciences, University of Illinois at Urbana-Champaign, Urbana, IL 61801, USA; shdi222@uky.edu (S.H.D.);; 3Department of Anesthesiology, The Center for the Study of Itch and Sensory Disorders, Washington University School of Medicine, St. Louis, MO 63110, USAyonghuizhao@wustl.edu (Y.Z.); 4Department Neuroscience Program, Division of Nutritional Sciences, and Carl R. Woese Institute for Genomic Biology, University of Illinois at Urbana-Champaign, Urbana, IL 61801, USA; 5Department of Pathology and Immunology, Washington University School of Medicine, Saint Louis, MO 63110, USA

**Keywords:** multiple sclerosis, disease activity, transient receptor potential vanilloid 4, TRPV4, microglia, inflammation, experimental autoimmune encephalitis, EAE, cuprizone

## Abstract

Microglia are found pathologically at all stages of multiple sclerosis (MS) lesion development and are hypothesized to contribute to both inflammatory injury and neuroprotection in the MS brain. Transient receptor potential vanilloid 4 (TRPV4) channels are widely expressed, play an important role as environmental sensors, and are involved in calcium homeostasis for a variety of cells. TRPV4 modulates myeloid cell phagocytosis in the periphery and microglial motility in the central nervous system. We hypothesized that TRPV4 deletion would alter microglia phagocytosis in vitro and lessen disease activity and demyelination in experimental autoimmune encephalitis (EAE) and cuprizone-induced demyelination. We found that genetic deletion of TRPV4 led to increased microglial phagocytosis in vitro but did not alter the degree of demyelination or remyelination in the cuprizone mouse model of MS. We also found no difference in disease in EAE following global or microglia-specific deletion of *Trpv4*. Additionally, lesioned and normal appearing white matter from MS brains exhibited similar *TRPV4* expression compared to healthy brain tissue. Taken together, these findings indicate that TRPV4 modulates microglial activity but does not impact disease activity in mouse models of MS, suggesting a muted and/or redundant role in MS pathogenesis.

## 1. Introduction

Multiple sclerosis is the most common chronic inflammatory disease of the central nervous system (CNS) affecting an estimated 2.8 million people worldwide [1]. Although the exact etiology of MS is unknown, it is considered an autoimmune disease in which lymphocytes and peripheral macrophages infiltrate the brain and spinal cord and cause multifocal inflammatory lesions [2]. Neuropathological data has demonstrated that acute lesions contain T lymphocytes, B lymphocytes, plasma cells, activated microglia, and macrophages [3]. While the pathogenesis of MS was historically hypothesized to be driven primarily by autoreactive T cells [4,5], a growing body of literature indicates a prominent role for microglia in MS [6].

Microglia have the potential to modulate MS disease activity in a variety of different ways via their diverse effector functions, including phagocytosis, antigen presentation, synaptic pruning, and the secretion of pro- and anti-inflammatory cytokines [7]. Microglia are found in high number in MS lesions and MS normal-appearing white matter (NAWM) [8]. In MS lesions, microglia are found surrounding the edge of active lesions and display an activated phenotype, suggesting they are important in plaque formation and tissue damage in MS [8]. Microglia have also been found within the MS brain in clusters termed “microglia nodules” that are associated with degenerating axons, stressed oligodendrocytes, and deposits of activated complement [9,10]. Microglia may also modulate the process of remyelination in MS. Phagocytosis of myelin debris by microglia promotes oligodendrocyte precursor cell differentiation and impacts the efficiency of remyelination [11]. Microglia-specific CX3CR1 knockout mice demonstrate reduced myelin clearance and decreased remyelination in the cuprizone mouse model [12]. Given this link between microglia and MS pathogenesis, a greater understanding of the cellular mechanisms underpinning microglial activation and function in MS will broaden our understanding of MS and may yield further therapeutic targets.

A growing body of literature indicates that the transient receptor potential (TRP) vanilloid 4 (TRPV4) channel modulates microglial activity in the context of neuroinflammation [13,14,15,16]. TRPV4 is a calcium-permeable, non-selective cation channel and a member of the TRP superfamily of cation channels [17]. TRPV channel dysfunction has been implicated in a variety of neurological disorders including MS [14]. TRPV1 stimulation has been found to induce an anti-inflammatory effect in microglia and MS patients with a TRPV1 single nucleotide polymorphism were found to have lower levels of pro-inflammatory TNF within their cerebrospinal fluid [18]. Within the TRPV receptor family, TRPV4 is the isoform most highly expressed in cortical microglia [19]. TRPV4 agonism has been found to increase neuroinflammation in an epilepsy mouse model [20] and increase blood–brain barrier permeability in endothelial cell culture [16]. On the other hand, TRPV4 antagonism has been found to decrease demyelination in the cuprizone mouse model [21,22]. Given the physiologic and therapeutic implications of TRPV4, further research into its role in MS is warranted. 

We utilized *Trpv4* knockout mice to explore the impact of *Trpv4* deletion on neuroinflammation and demyelination observed in models of MS. We utilized both in vitro and in vivo model systems and analyzed TRPV4 expression in the brain tissue of MS patients. Our in vitro assay demonstrated increased microglia phagocytosis following *Trpv4* deletion but our experimental autoimmune encephalitis (EAE) and cuprizone animal models showed no impact of global or microglia-specific *Trpv4* deletion on disease severity or demyelination, respectively. We also observed no change in TRPV4 expression in MS lesions or NAWM compared to healthy control tissue. Taken together, these findings indicate that loss of TRPV4 increases microglia phagocytosis in vitro but that TRPV4 is unlikely to play a direct role in MS pathogenesis.

## 2. Results

### 2.1. Trpv4 Deletion in Microglia Leads to Increased In Vitro Phagocytosis but Does Not Alter the Degree of Demyelination, Remyelination, or Microgliosis following Cuprizone Treatment

Microglia are hypothesized to facilitate repair following demyelination in MS via phagocytosis of myelin debris [12]. To determine if Trpv4 deletion alters microglial phagocytosis, we performed an in vitro phagocytosis assay on microglia isolated from global Trpv4 knockout (KO), or Trpv4^KO^, mice. Loss of TRPV4 led to a 7.3% increase in the percentage of microglia performing phagocytosis (23.50% of Trpv4^KO^ microglia performed phagocytosis and 16.18% of wild-type (WT) microglia performed phagocytosis (95% CI, 0.15–13.04%), *p* < 0.05). Trpv4^KO^ microglia that engaged in detectable phagocytosis also contained a greater number of beads per cell compared to WT microglia (3.60 per microglia in KO microglia (95% CI, 2.88–4.50) vs. 3.00 per microglia in WT microglia (95% CI, 2.04–3.95), *p* < 0.05) (Figure 1). 

Given the observed increase in phagocytosis in Trpv4^KO^ microglia, we next evaluated the impact of global Trpv4 deletion using the cuprizone mouse model to determine if this change translated to in vivo differences in demyelination and/or remyelination (Figure 2). Despite the difference in microglial phagocytosis in vitro, there was no difference in the degree of myelin loss following acute demyelination (6 weeks of cuprizone administration) in global Trpv4^KO^ mice (Figure 3). No difference in myelin basic protein (MBP) mean fluorescence intensity (MFI) was found when comparing Trpv4^KO^ and WT mice (94.20 in Trpv4^KO^ mice (95% CI, 21.74–49.17) vs. 104.10 in WT mice (95% CI, 23.83–58.96), *p* = 0.64). Similarly, myelin scoring in the splenium of the corpus callosum was found to be similar in Trpv4^KO^ and WT mice (1.57 in Trpv4^KO^ mice (95% CI, 1.07–2.06) vs. 1.25 in WT mice (95% CI, 0.86–1.63), *p* = 0.23) (Figure 4). 

To more specifically investigate the impact of Trpv4 deletion on microglia in the cuprizone mouse model, we utilized mice with microglial-specific deletion of Trvp4, termed Trpv4^CX3CR1-KO^ mice, in both acute demyelination (6 weeks of cuprizone feed) and chronic demyelination/remyelination (Figure 2). We found no impact of microglia-specific loss of Trpv4 in the myelin content of the splenium of the corpus callosum after acute demyelination determined via MBP MFI (81.13 MFI in Trpv4^CX3CR1-KO^ mice (95% CI, 59.58–110.00) vs. 111.12 MFI in WT mice (95% CI, 59.19–164.00), *p* = 0.20). Further, there was no difference in MBP MFI at the chronic demyelination/remyelination stage between Trpv4^CX3CR1-KO^ and WT mice (65 MFI in Trpv4^CX3CR1-KO^ mice (95% CI, 43.73–86.16) vs. 81.25 MFI in WT mice (95% CI, 59.22–103.30), *p* = 0.23). These observations were also reflected by myelin scoring measures (acute demyelination: 2.56 in Trpv4^CX3CR1-KO^ mice (95% CI, 2.18–2.95) vs. 3.00 in WT mice (95% CI, 2.06–3.93), *p* = 0.2507, chronic demyelination/remyelination: 1.75 in KO mice (95% CI, 1.36–2.14) vs. 2.00 in WT mice (95% CI, 2.00–2.00), *p* = 0.26) (Figure 4). We then quantified the extent of microgliosis and found no difference in Iba1 staining within the splenium of the corpus callosum in either global Trpv4^KO^ or the microglia-specific Trpv4^CX3CR1-KO^ mice compared to WT controls (Figure 5).

These results indicate that Trpv4 deletion causes increased microglia phagocytosis in vitro but neither global nor microglia-specific Trpv4 deletion alters the degree of myelin loss in vivo in the cuprizone mouse model. Additionally, Trpv4 deletion does not alter the degree of microgliosis following cuprizone-mediated demyelination.

### 2.2. Global and Microglia-Specific Trpv4 Deletion Does Not Impact EAE

Microglia are hypothesized to modulate neuroinflammation in MS and EAE via antigen presentation and pro-inflammatory cytokine secretion, potentially contributing to disease activity [7,23,24]. Given this immunomodulatory role of microglia, we explored the impact of TRPV4 on the EAE mouse model. We utilized global Trpv4^KO^ mice and microglia-specific Trpv4 knockouts (Trpv4^CX3CR1-KO^ mice). We observed no change in EAE disease onset or severity in global Trpv4^KO^ mice and microglia-specific Trpv4^KO^ mice compared with WT mice. To account for any impact of Trpv4 on severity of disease alone, we performed a separate set of analysis excluding mice that did not develop EAE. The results were similar, with no statistical differences between EAE severity in Trpv4^KO^ vs. WT mice (Figure 6A) or Trpv4^CX3CR1-KO^ vs. WT mice (Figure 6B). In our global Trpv4^KO^ cohort, there was an absolute increase in disease score between days 15–20 that subsequently dropped to the level of the disease score of WT mice. When this specific time interval was analyzed for statistical significance via Mann–Whitney U testing, it was found to be nonsignificant. These results demonstrate that neither global nor microglia-specific loss of Trpv4 alters disease activity in EAE.

### 2.3. Expression of TRPV4 in MS Lesions, MS Normal-Appearing White Matter, and Control Tissue

We next analyzed MS brain tissue to determine if TRPV4 is differentially expressed in MS brain tissue. We examined expression of TRPV4 in chronic inactive lesions, active lesions, and NAWM in MS and compared it to NAWM from healthy controls (Table 1). To determine the amount of TRPV4 gene expression in MS lesions and MS normal appearing white matter, we isolated RNA from the brain tissue of MS patients and quantified the level of TRPV4 expression via quantitative PCR. We found no difference in TRPV4 expression between chronic inactive lesions, active lesions, MS NAWM and control tissue (Figure 7). We also found no difference overall in TRPV4 gene expression between MS and non-MS (control) brain tissue, although there was a trend towards increased TRPV4 gene expression in MS brain tissue that did not achieve statistical significance. These results indicate that TRPV4 gene expression is similar between both MS lesions, MS NAWM, and healthy control tissue.

## 3. Discussion

Our results indicate that loss of TRPV4 increases microglia phagocytosis in vitro but does not alter the degree of demyelination, remyelination or microgliosis in the cuprizone model of MS. Similarly, in vivo loss of TRPV4, globally or selectively in microglia, did not affect disease severity in the active EAE model of MS. Consistent with these in vivo observations, no difference in *TRPV4* gene expression was observed between MS brain tissue and control tissue, indicating that *TRPV4* is not differentially expressed in the MS brain. Overall, these findings indicate that TRPV4 modulates microglia activity, but this effect is unlikely to have an impact on the pathogenesis of MS. 

Our study employed microglia-specific *Trpv4* genetic knockouts to study the effect of TRPV4 in microglia in vitro and on the MS mouse models of cuprizone and EAE. Our in vitro findings of increased phagocytosis following TRPV4^KO^ are consistent with prior research demonstrating that TRPV4 activation decreased LPS-induced microglia activation [25]. Our results are consistent with the lack of an effect of global TRPV4^KO^ on EAE observed previously [16].

There are many possible explanations for the seemingly opposing findings of increased microglia phagocytosis in vitro without an apparent effect of TRPV4 deletion in MS mouse models. Microglia are hypothesized to play a multifaceted role in MS, performing both pro-inflammatory [8] and anti-inflammatory [26] roles throughout the MS disease course. Additionally, microglia have been found to both enhance [26,27] and impair [28] remyelination in MS depending on their cytokine expression profile and phenotype. Given the potential dual role of microglia in MS, loss of TRPV4 may lead to equal and opposite effects on microglia causing a net neutral result. For example, if TRPV4 increased microglia phagocytosis in MS, as we observed in vitro, but also increased expression of pro-inflammatory cytokines, this would lead to increased myelin clearance, subsequently decreasing inflammation and promoting remyelination. However, this would simultaneously lead to increased neuroinflammation through the secretion of pro-inflammatory cytokines, leading to decreased remyelination. The net result would be a comparable level of disease pathology. 

Another possible explanation is that TRPV4 modulation alters the phenotype of homeostatic microglia but does not impact microglia in the context of cuprizone-mediated demyelination, EAE, and MS. This has been observed experimentally with endothelial cells in which inhibition of TRPV4 under homeostatic conditions leads to increased endothelial cell permeability and increased blood–brain barrier permeability but in states of inflammation, such as EAE, this effect does not occur [16]. A final explanation is that the effect of increased microglia TRPV4-mediated phagocytosis is negligible in the larger context of MS, where other disease processes predominate. 

Our findings conflict with a previous study by Liu et al. 2018 that found that intracerebroventricular infusion of a TRPV4 antagonist in mice receiving cuprizone diminished the degree of demyelination and reduced the number of astrocytes and microglia in the corpus callosum [22]. This difference in outcome may be related to the difference in mechanism of TRPV4 deletion employed in our respective studies. Liu et al. performed transient TRPV4 antagonism using intracerebroventricular infusion of a TRPV4 antagonist. In contrast, in our experiments a constitutive global and microglia-specific TRPV4 knockout genetic approach were used to assess in vivo function. This difference could result in temporal, spatial, and cell-specific differences between our two models which could then translate to clinical differences. Possible disparities attributable to the mechanism of TRPV4 loss of function is supported by prior research which demonstrated that pharmacologic TRPV4 inhibition with an TRPV4 antagonist resulted in changes to microglia morphology and motility but constitutive, global deletion of *Trpv4* had no effect on microglia [29]. Additionally, CNS delivery of a TRPV4 antagonist produces a localized effect in the CNS. In contrast, our global knockouts have systemic effects which may lead to alterations in cells within the periphery that may negate the neuroprotective effects occurring locally within the CNS. For example, a previously described shift in peripheral macrophages towards a pro-inflammatory phenotype in TRPV4^KO^ mice [30] could lead to a higher infiltration of macrophages in the CNS and a higher degree of neuroinflammation. This infiltration of pro-inflammatory myeloid cells into the CNS could obviate local neuroprotective effects from TRPV4 loss in microglia. Finally, our microglia-specific Trpv4^CX3CR1-KO^ selectively deletes *Trpv4* in CNS microglia, which results in primary changes only to microglia, whereas local pharmacologic TRPV4 antagonism causes changes in all CNS cells expressing TRPV4. Therefore, if local antagonism achieves a therapeutic benefit by blocking TRPV4 on CNS cells other than microglia, then this benefit would not be observed in microglia-specific Trpv4^CX3CR1-KO^ mice. 

Our research has several limitations, including narrow in vitro experimental approaches. We chose to focus our experiments primarily on the effect of TRPV4 loss in mouse models of MS and therefore did not conduct further in vitro assays using microglia from TRPV4^KO^ mice. However, further in vitro assays may help elucidate the mechanism by which loss of TRPV4 induces changes in microglia and demonstrate the broader effects of these changes. Additionally, a lack of techniques with greater resolution for quantifying myelination (e.g., 3D electron microscopy) could be one limitation in interpreting the extent to which TRPV4 function in microglial affects demyelination and remyelination in vivo. Nevertheless, we chose to quantify myelination in our cuprizone mice via immunohistochemical staining with MBP and solochrome cyanine which demonstrated consistency in staining patterns across the two stains, increasing our confidence in the results (Appendix A).

Overall, the lack of any significant change in myelin content, microgliosis, or disease severity in the cuprizone and EAE mouse models with global and microglia-specific loss of TRPV4 indicates that TRPV4 is unlikely to play a prominent role in MS pathogenesis. This is supported by a lack of differential gene expression of TRPV4 in the brain tissue of MS patients. It is possible that the direct application of a TRPV4 antagonist to the CNS results in a differential effect compared to the constitutive deletion of *Trpv4*, leaving open the possibility of a therapeutic potential for transient TRPV4 antagonism. This warrants further evaluation through experiments that compare the functional and morphological differences between transient and constitutive loss of TRPV4 in microglia as well as other CNS cells such as endothelial cells. We cannot rule out the possibility that a toxic gain of function of TRPV4 could contribute to disease pathogenesis in MS. Our study specifically examined *Trpv4* deletion through animal modeling and in vitro experiments and therefore cannot comment specifically on the impact of toxic gain of function of TRPV4. However, our human data did not demonstrate an increased expression of TRPV4 within MS tissue, making toxic gain of function of TRPV4 less likely. Cell-based or in vivo experiments that evaluate the phagocytosis of myelin in microglia deficient in, or over-expressing TRPV4 could provide further information about how TRPV4 modulates myelin-specific phagocytosis. These experiments may offer insight into the mechanisms by which TRPV4 mediates functional changes in neuroinflammation. 

## 4. Materials and Methods

### 4.1. Microglia Cultures

Primary microglia were obtained from mixed cultures prepared from the hippocampi and cerebral cortices of TRPV4^KO^ mice and wild type C57BL/6J mice at postnatal day (P)1-3 as previously described [31]. Microglial cells were isolated by shaking flasks for 45 min at 230 RPM on day 10 after plating. Cells were then seeded on poly L-ornithine (Sigma; St. Louis, MO, USA) pre-coated wells at a density of 1.53105 cell/mL in DMEM containing 20% heat-inactivated fetal bovine serum. For biological replication, ten different mice and two sets of cultures were used in each experimental group.

### 4.2. Microglia Phagocytosis Assay

Red fluorescent latex beads (Sigma, L2778; St. Louis, MO, USA) were pre-opsonized with FBS for one hour and then co-cultured with microglia derived from Trpv4 −/− mice (global Trpv4^KO^) and wild type microglia in DMEM using 100,000 microglia per sample. Microglia were incubated for three hours with the latex beads and then fixed and stained for Iba1 (Wako; Richmond, VA, USA). Microglia were analyzed using a Zeiss LSM880 Confocal Laser Scanning Microscope (Carl Zeiss Microscopy; White Plains, NY, USA) and the number of internally phagocytosed fluorescent beads per microglia were counted. For technical replication, 10 separate microscopic fields with at least 500 microglia total were quantified per cell culture.

### 4.3. Mice

All animal experiments followed the guidelines of the National Institutes of Health (NIH, New York, NY, USA), and all animal protocols were approved by the Institutional Animal Care and Use Committees (IACUC) of Washington University in St. Louis. Mice were housed in a temperature- and humidity-controlled animal facility employing a 12 h light/dark cycle with food and water available ad libitum. Global Trpv4^KO^ mice [32] and Trpv4^f/f^; CX3CR1^CreERT2^ mice (microglia-specific Trpv4^CX3CR1-KO^ mice) were bred in-house (HH lab) [21,33]. Microglia-specific Trpv4^CX3CR1-KO^ mice were treated with 100 mg/kg of tamoxifen administered via oral gavage for 5 consecutive days to induce Cre-Lox recombination. Genotypes were determined by PCR from the DNA extracted from tail tips. Trvp4 flox alleles were genotyped by the primer set (5′-CCAAGACAGGCAAGATCGGG-3′; 5′-CTGACTGATGGAGGTTGGGT-3′). Genotyping CX3CR1^CreERT2^ mice utilized the following primers: 5′-CCAAGACAGGCAAGATCGGG-3′; 5′-CTGACTGATGGAGGTTGGGT-3′. The recombined *Trpv4* allele in microglia-specific Trpv4^CX3CR1-KO^ mice following tamoxifen administration was genotyped using primers (5′-ACCCTCATTTTGGTCCATCC-3′; 5′-CTGACTGATGGAGGTTGGGT-3′). Microglia were isolated from microglia-specific Trpv4^CX3CR1-KO^ mice via FACS sorting and genotyped to confirm successful recombination (Appendix A). Both females and males were used in this study. WT litter mates served as controls for all animal experiments.

### 4.4. EAE Induction and Clinical Score Assessment

Active EAE was induced as previously described [34]. Briefly, mice were subcutaneously injected with 200 μg of MOG_35–55_ peptide (Genscript, RP10245) emulsified in incomplete Freud’s adjuvant (Sigma, F5506; St. Louis, MO, USA) with mycobacterium tuberculosis H37Ra (BD Difco, BD 231141; Pittsburgh, PA, USA) and intraperitoneally injected with 200 ng of Pertussis Toxin (List Biological Laboratories, Campbell, CA, USA). All EAE mice were monitored daily and scored using a clinical scale from 0 to 5 (0: no abnormality; 1: limp tail; 2: limp tail and hind leg weakness; 3: limp tail and complete paralysis of hind legs; 4: hind leg and partial front leg paralysis; 5: moribund). For biological replication, the TRPV4 global knockout cohort contained 15 mice, the CX3CR1 Trpv4 knockout cohort contained 13 mice, and the wild type cohorts each contained 10 mice.

### 4.5. Cuprizone-Induced Acute Demyelination and Chronic Demyelination/Remyelination Models

Cuprizone is a copper-chelating mitochondrial toxin that causes oligodendroglial cell death and demyelination by an inhibition of complex IV of the mitochondrial respiratory chain and induces megamitochondria in the mouse brain. To induce acute demyelination in the CNS, global TRPV4^KO^ and microglia-specific Trpv4^CX3CR1-KO^ mice, standard 0.3% cuprizone diet in chow (Sigma; St. Louis, MO, USA) was provided for a period of six weeks as previously described [35]. Another group of microglia-specific Trpv4^CX3CR1-KO^ mice were fed a standard 0.3% cuprizone diet in chow (Sigma; St. Louis, MO, USA) for 12 weeks followed by two weeks of regular feed (Gateway lab supply, St Louis, MO, USA) to model chronic demyelination and subsequent remyelination as previously described [36]. The body weight of all mice was monitored once a week. For biological replication, the global TRPV4^KO^ cohort that underwent acute demyelination contained 8 mice and the wild type cohort contained 8 mice. For the microglia-specific Trpv4^CX3CR1-KO^ mice, the knockout cohort that underwent acute demyelination contained 8 mice and the wild type cohort contained 8 mice, and the knockout cohort that underwent chronic demyelination and remyelination contained 15 mice and the wild type cohort contained 8 mice.

### 4.6. Immunohistochemistry

The mice were anesthetized and perfused transcardially with PBS and 4% PFA. The spinal cord and brain were dissected and post fixed in 4% paraformaldehyde (PFA) overnight at 4 °C, followed by equilibration in 30% sucrose. Subsequently, 20 μm cryosections were collected from brains at the splenium of the corpus callosum and stored at −20 °C. Sections were incubated with primary antibodies overnight at 4 °C (MBP: Abcam; Eugene, Oregon Iba1: Wako; Richmond, VA, USA) and washed and incubated with fluorescently conjugated secondary donkey anti-rabbit IgG 488 (Thermofisher; Waltham, MA, USA) or donkey anti-rat IgG 647 (Invitrogen; Rockford, IL, USA) antibodies for 1 h at room temperature. Images were collected and analyzed using a Zeiss LSM880 Confocal Laser Scanning Microscope. 

### 4.7. Myelin Content Quantification

Myelin content was quantified by both mean fluorescent intensity and blinded myelin scoring. Serial sections of the corpus callosum at approximately −2.2 to −2.4 mm posterior from Bregma were collected for each mouse, stained with MBP (Abcam; Eugene, OR, USA), and analyzed using a Zeiss LSM880 Confocal Laser Scanning Microscope. For technical replication, two serial 30 um corpus callosum slices at approximately −2.2 to −2.4 mm posterior from Bregma were collected and stained for each mouse. Mean fluorescent intensity was calculated after outlining the corpus callosum in each image and averaging the signal intensity of both slices. Myelin content in each image was also scored by a blinded rater from an independent lab as described previously [37], where a score of 0 indicates no demyelination detected, 1 indicates mild demyelination, 2 moderate demyelination, 3 indicates severe demyelination, and 4 indicates total demyelination.

### 4.8. Solochrome Cyanine Staining

For myelin staining, sections were air-dried and carried through the following sequence: 95% ethanol, 95% ethanol/formaldehyde; 95% ethanol, 70% ethanol, dH_2_O, then to the solochrome staining solution (80mL of Solochrome stock solution (1.5% solochrome, 2.5% *v*/*v* sulfuric acid), 320 mL dH_2_O,100 mL 4% ferric ammonium sulfate) at room temperature. Sections were then rinsed with tap water, differentiated in potassium ferricyanide-sodium borate, and rinsed in tap water again. Differentiated sections were counterstained lightly with neutral red, dehydrated, cleared in xylene and cover slipped. For technical replication, three serial 30 um corpus callosum slices at approximately −2.2 to −2.4 mm posterior from Bregma were collected and stained.

### 4.9. Quantification of TRPV4 Gene Expression in MS Brain Tissue

Fresh frozen tissue blocks were obtained from the John L. Trotter MS tissue biorepository at Washington University from the brain tissue of MS patients (demographics listed in Table 1). MS lesions and healthy control tissue were obtained from periventricular cerebral white matter. RNA from tissue sections was extracted with the RNAeasy kit (QIAGEN; Germantown, MD, USA) according to the manufacturer’s instructions. cDNA was created using SuperScript IV Reverse Transcriptase according to the manufacturer’s instructions. Quantitative real-time PCR was carried out using TaqMan Fast Advanced Master Mix (ThermoFisher; Waltham, MA, USA) on an ABI 7900HT instrument. *GAPDH* was used as the normalization control and the DDCT method was used to quantify gene expression. 

### 4.10. Statistical Analysis

All statistical analyses were performed using GraphPad Prism software, V 9.0 (GraphPad Software, La Jolla, CA, USA). Normality was tested by the Shapiro–Wilk test and data are shown as the mean and standard deviation. Statistical difference in microglia phagocytosis was determined using a nested Student’s *t*-test. Difference in EAE disease course was determined using the Mann–Whitney U test. Difference in *TRPV4* gene expression in MS and control brain tissue was determined using one-way ANOVA. In all other experiments, comparison of mean values was conducted using the Student’s *t*-test. For all statistical tests, two-sided *p*-values of 0.05 or less were considered significant. 

## Figures and Tables

**Figure 1 ijms-24-17097-f001:**
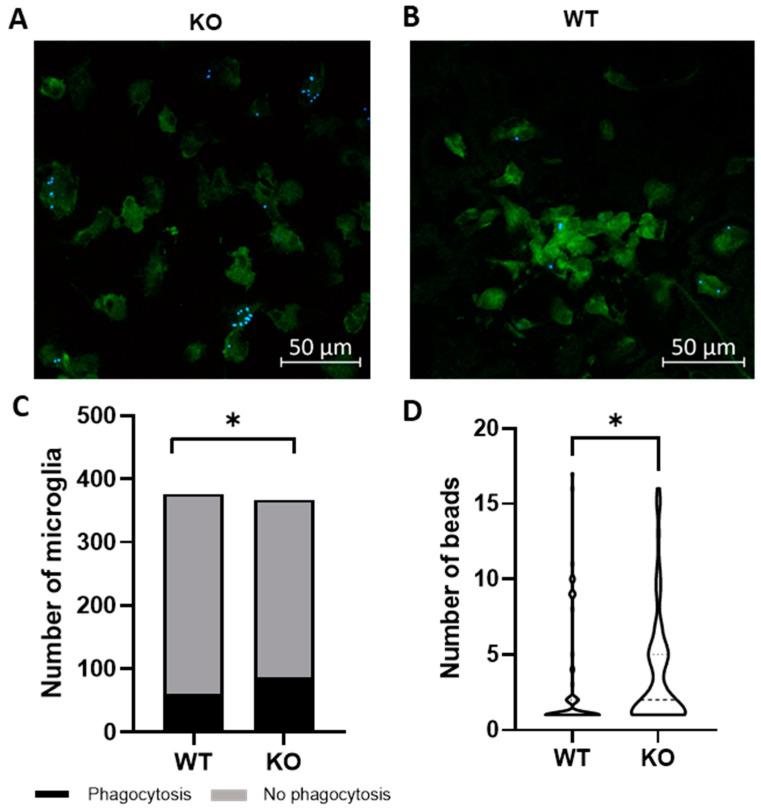
Global Trpv4^KO^ leads to increased microglia phagocytosis of fluorescent beads in vitro. (**A**,**B**) Microglia were cultured from P1-P3 global TRPV4^KO^ (**A**) and WT (**B**) mice and incubated with fluorescent latex beads for three hours to measure microglia phagocytosis. Green = Iba1, Blue = fluorescent latex beads. (**C**) Quantification of the number of microglia that phagocytose fluorescently labelled latex beads in vitro. (**D**) Quantification of the number of beads phagocytosed per microglia. Only microglia with at least one internalized bead were analyzed (*n* = 64 microglia in WT cohort and *n* = 79 in KO cohort). Scale bars indicate 50 µm. Statistical differences determined using a nested Student’s *t*-test. Microglia were derived from ten different mice per experimental group. Two sets of cultures were obtained. * *p* < 0.05. WT = wild type microglia, KO = global TRPV^KO^ microglia.

**Figure 2 ijms-24-17097-f002:**
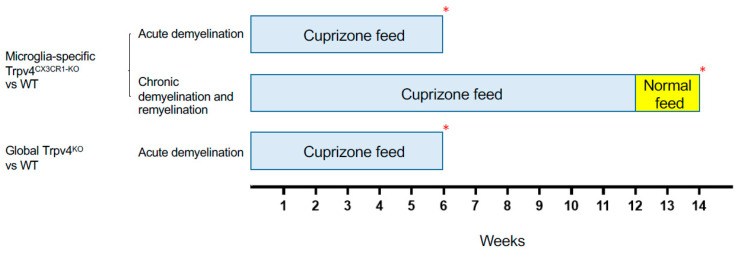
Schematic of the cuprizone protocols used to induce demyelination and remyelination in different experimental cohorts. Mice in the acute demyelination cohorts were fed cuprizone for 6 weeks and were then sacrificed and analyzed. Mice in the chronic demyelination and remyelination cohort were fed cuprizone for 12 weeks, normal feed for 2 weeks, and were then sacrificed and analyzed. Each experimental cohort was compared to a WT cohort that underwent the same protocol. The red asterisk (*) indicates the time point at which the mice were sacrificed. WT = wild type.

**Figure 3 ijms-24-17097-f003:**
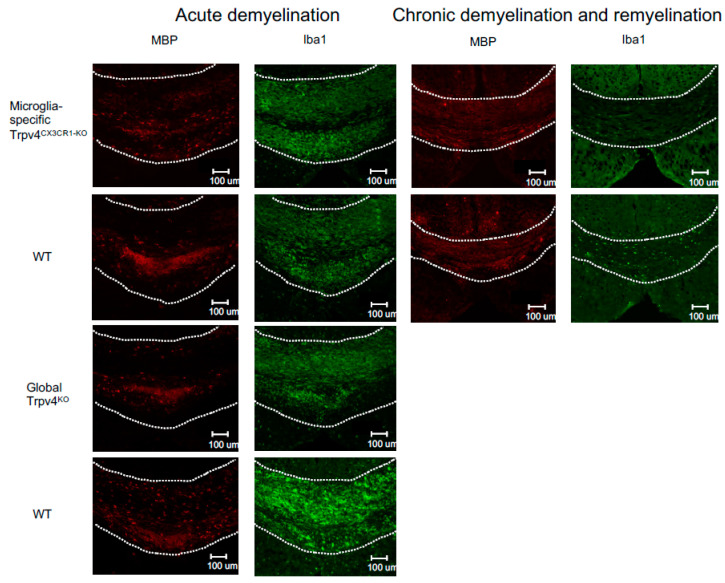
Representative immunofluorescent images of the splenium of the corpus callosum from microglia-specific Trpv4^CX3CR1-KO^ and global Trpv4^KO^ mice treated with cuprizone. Mice in each group were treated with either 6 weeks of cuprizone (acute demyelination protocol) or 12 weeks of cuprizone and 2 weeks of regular feed (chronic demyelination and remyelination protocol). All images were taken from the same reference point in the splenium of the corpus callosum. MBP staining (first and third column) visualizes myelin (in red). Iba1 staining (second and fourth column) visualizes microglia density (in green). The dotted white line demarcates the borders of the splenium of the corpus callosum. Scale bars indicate 100 µm. WT = wild type.

**Figure 4 ijms-24-17097-f004:**
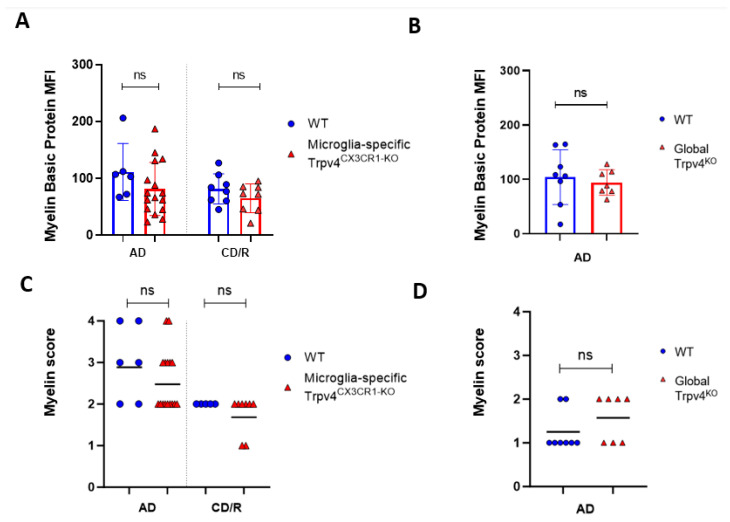
Global or microglial-specific loss of Trpv4 does not alter the degree of myelination following cuprizone administration. (**A**) Quantification of myelin content measured via myelin basic protein mean fluorescent intensity (MFI) after a 6-week acute demyelination and 14-week chronic demyelination/remyelination protocol in microglial-specific Trpv4^KO^ mice. (**B**) Quantification of myelin content via myelin basic protein MFI after a 6-week acute demyelination protocol in global Trpv4^KO^ mice. (**C**) Quantification of the myelin content via myelin scoring using the confocal images analyzed in (**A**). (**D**) Quantification of the myelin content via myelin scoring using the confocal images analyzed in (**B**). Myelin content was scored by a blinded rater as described previously (23) where a score of 0 indicates no demyelination detected, 1 indicates mild demyelination, 2 moderate demyelination, 3 indicates severe demyelination, and 4 indicates total demyelination. Statistical differences were determined using a Student’s *t*-test. ns = non-significant. AD = acute demyelination protocol, CD/R = chronic demyelination/remyelination protocol, MFI = mean fluorescent intensity.

**Figure 5 ijms-24-17097-f005:**
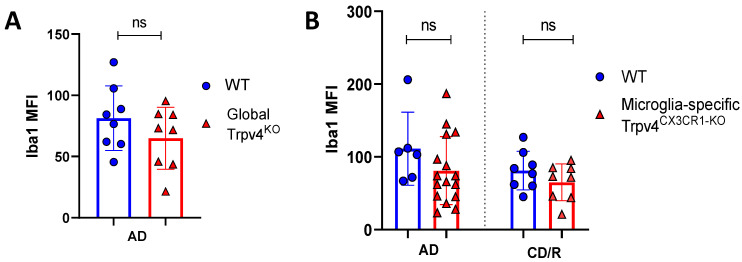
Global or microglial-specific loss of *Trpv4* does not alter the degree of microgliosis following cuprizone administration. (**A**) Quantification of the degree of microgliosis via mean fluorescent intensity after a 6-week demyelination protocol in global Trpv4^KO^. (**B**) Quantification of the degree of microgliosis using Iba1 mean fluorescent intensity (MFI) measured via confocal microscopy after a 6-week acute demyelination and 14-week chronic demyelination/remyelination protocol in microglial-specific Trpv4^CX3CR1-KO^ mice. MFI was calculated using confocal images of fixed coronal sections of the splenium of the corpus callosum stained with Iba1. Statistical differences were determined using a Student’s *t*-test. MFI = mean fluorescence intensity; ns = non-significant; AD = acute demyelination protocol; CD/R = chronic demyelination/remyelination protocol.

**Figure 6 ijms-24-17097-f006:**
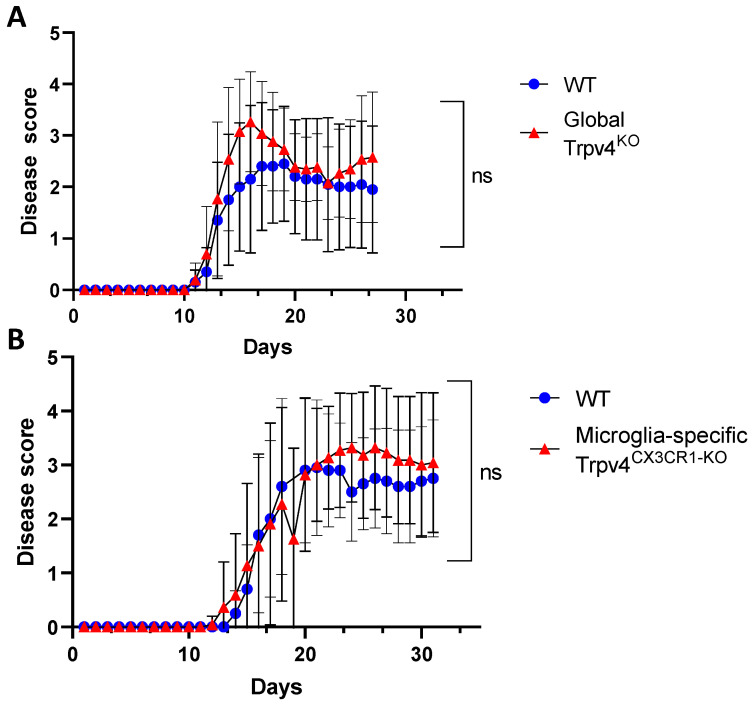
Microglia-specific Trpv4^CX3CR1-KO^ and global Trpv4^KO^ does not alter EAE disease severity. (**A**) EAE disease severity in wild type and CX3CR1 Trpv4 knockout mice. The wild type cohort contained 10 mice and the TRPV4 global knockout cohort contained 15 mice. Statistical differences were determined using the Mann–Whitney U test. (**B**) EAE disease severity in wild type and microglia-specific Trpv4^CX3CR1-KO^ mice. The wild type cohort contained 10 mice and the CX3CR1 Trpv4 knockout cohort contained 13 mice. After the initial data analysis, the data was reanalyzed excluding mice that did not develop EAE and demonstrated similar results. The graphs above depict the results from the reanalysis. ns = non-significant, WT = wild type.

**Figure 7 ijms-24-17097-f007:**
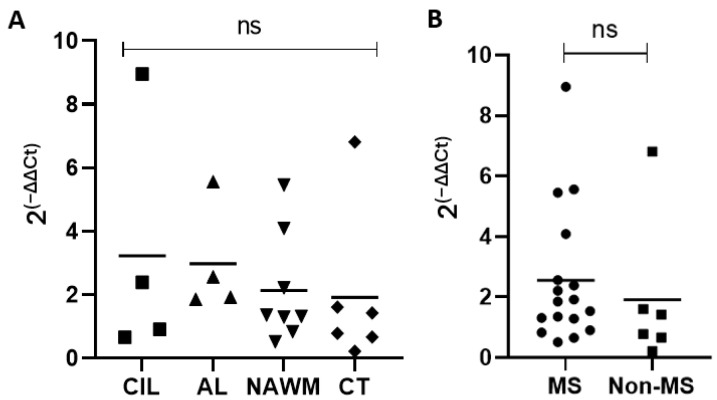
TRPV4 gene expression does not differ between MS lesions, MS normal appearing white matter, and healthy control tissue. (**A**) TRPV4 gene expression in chronic inactive MS lesions, active MS lesions, MS NAWM, and healthy control brain tissue. (**B**) Combined analysis of TRPV4 gene expression in MS lesions + MS NAWM (MS) and healthy control brain tissue (Non-MS). Y axis for both graphs indicates relative fold expression change in TRPV4 compared to GAPDH. Statistical differences determined using a Student’s *t*-test. ns = non-significant. CIL = chronic inactive lesions, AL = active lesions, NAWM = MS normal appearing white matter, CT = control tissue.

**Table 1 ijms-24-17097-t001:** Demographic information for MS cases and controls. Demographic information on the multiple sclerosis cases and healthy control cases from which brain tissue was sampled for measurement of TRPV4 gene expression via real-time PCR.

Sex/Age	MS Subtype	PMI (Hours)	Cause of Death	Lesion Type
M/60	SPMS	9	Respiratory failure	Active
F/45	PPMS	4	Pneumonia	Chronic active
F/50	SPMS	20	Unknown	Active, chronic inactive
M/35	PPMS	10	Unknown	Chronic inactive
F/54	SPMS	8	Pneumonia	Chronic inactive
F/69	PPMS	5	Metastatic colon cancer	Active
F/79	RRMS	8	Pulmonary edema	NAWM, chronic inactive
F/95	SPMS	9	Pulmonary edema	NAWM
F/41	RRMS	12	Complications from T1D	NAWM
F/54	SPMS	7	Pneumonia	Chronic inactive
F/86	PPMS	12	Cardiac arrest	NAWM
F/66	Unknown	29	Unknown	NAWM
F/39	HC	4	CNS lymphoma	NA
F/56	HC	18	Myocardial infarction	NA
F/69	HC	43	Sepsis	NA
M/41	HC	24	Heart failure	NA

PMI = post-mortem interval; T1D = Type 1 diabetes; NAWM = Normal appearing white matter; NA = not applicable. PPMS = primary progressive multiple sclerosis, SPMS = secondary progressive multiple sclerosis, RRMS = relapsing–remitting multiple sclerosis.

## Data Availability

The data presented in this study are openly available on request from the corresponding author at synapse.org (syn52309481).

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
