# Peer review of "Transient Receptor Potential Vanilloid 4-Dependent Microglial Function in Myelin Injury and Repair"

_ijms, 2023, doi:10.3390/ijms242317097_

Round 1
Reviewer 1 Report
Comments and Suggestions for Authors
In order to reveal the relationship of Transient Receptor Potential Vanilloid 4-Dependent Microglial Function in Myelin Injury and Repair, the authors analized the TRPV4 deleted in vitro experiments.
The authors reached out to the conclusion that genetic deletion of TRPV4 led to increased microglial phagocytosis in vitro but did not alter the degree of demyelination or remyelination in the cuprizone mouse model of MS.
And also the authors found no difference in disease in EAE following global or microglia-specific deletion of Trpv4.
But I am still skeptical for the deletion of TRPV4. In the supplement data, WesternBlot data was shown. It seems like totally very thin bands on the membrane. And also, any negative feedback or alternative signal could contribute to microglial phagocytosis etc, alternatively. To clarify the concern, I recommend that the atuhors shoul reconfirm the evidence of the Trpv4 deletion. If this is OK, this manuscript could be published in the journal.
Reviewer 2 Report
Comments and Suggestions for Authors
The study explores the role of TRPV4 in MS pathophysiology by using targeted and global deletion approaches. The study design is methodical and rigors. Their findings indicate that knocking out TRPV4 has no impact on the myelination process in both chronic and acute demyelination models. Overall, the study is well-presented and has a logical flow. Moreover, negative results are as important as positive findings and have the same scientific value when the study is well-conducted.
There are a number of points that if addressed will improve the manuscript:
A. Introduction:
Line 75: The authors need to revise the cited reference. Ref 21 describes a study on the role of TRPV4 in a skin condition. The condition is not related to MS and there is no description of cuprizone application in the cited study.
B. Methodology:
1) Description of how the technical and biological repeats were defined for each experiment is missing
2) Line 346: which cell-type specific marker was used to sort microglia?
3) EAE induction and clinical score assessment: How long does it take for the animal to manifest the diseases after receiving the manipulation? did the authors allow sufficient time for the disease to develop and the symptoms to appear?
4) Myelin scoring: the myelin scoring method which is conducted as previously described (Steelman, et al..) is a bit ambiguous. The percentage-based scoring system that the authors used may imply that a semi-quantitative method was applied. How exactly did the rater designate the percentage range for each image/brain slice? did they for example calculate the fraction of area with faded staining (for demyelination) or area with strong staining (for intact myelination) to the total area demarcated within the borders of the splenium of the corpus callosum. Instead of using numerical values to score the myelination status the authors can consider using descriptive terms such as (No demyelination detected, mild demyelination, moderate and severe).
5) Quantification of TRPV4 gene expression in MS brain tissue: type of sorted cells and used marker is not indicated here. Also, which brain region and subregion the tissue was extracted from?
C. Results:
1) Figure 2: why the authors have not included a control group (WT-no treatment) this will provide a baseline of the myelination signal for the reader to use a reference when examining the acute and chronic demyelination states. This is useful for two reasons; (1) to ensure that the demyelination protocol is successful (2) to ensure that the remyelination protocol restored the myeline to the baseline level.
2) Figure 3: images representative of remyelination are not presented why?
3) Figure 4: It is not clear if the data points in each plot for each condition represent slices from the same animal or different animals, it is important to provide a clear description of what was considered as technical replicates and biological replicates for each experiment.
4) Line 160: the scoring of demyelination described here is to the decimal point precision as opposed to arbitrary percentage scoring as previously described (Figure 4 legend), can the authors explain?
5) Figure 6A: there seems to be a rise in disease score for the days between 15-20 before it drops, can the authors comment on that.
6) Table 1: identification of MS subtypes listed in the table is missing.
D. Discussion:
The approaches used to study the role of TRPV4 in demyelination here and in Liu et al2018, are based on a loss of function mechanism of TRPV4, the authors can discuss if the possibility of toxic gain of function mechanism of TRPV4 is a plausible theory.
Round 2
Reviewer 1 Report
Comments and Suggestions for Authors
I think the author's answered my question on the revised version. Indeed, I can recommend to publish this manuscript in the IJMS.